# Distinct Neurodegenerative Pathways in Two NBIA Subtypes: Inflammatory Activation in *C19orf12* but Not in *PANK2* Mutation Carriers

**DOI:** 10.3390/cells14221801

**Published:** 2025-11-17

**Authors:** Marta Skowrońska, Agnieszka Cudna, Barbara Pakuła, Magdalena Lebiedzińska-Arciszewska, Justyna Janikiewicz, Aneta M. Dobosz, Patrycja Jakubek-Olszewska, Agata Wydrych, Maciej Cwyl, Agnieszka Dobrzyń, Mariusz R. Więckowski, Iwona Kurkowska-Jastrzębska

**Affiliations:** 12nd Department of Neurology, Institute of Psychiatry and Neurology, 02-957 Warsaw, Poland; mskowronska@ipin.edu.pl (M.S.); acudna@ipin.edu.pl (A.C.); 2Laboratory of Mitochondrial Biology and Metabolism, Nencki Institute of Experimental Biology, Polish Academy of Sciences, 02-093 Warsaw, Polandm.lebiedzinska@nencki.edu.pl (M.L.-A.);; 3Laboratory of Cell Signaling and Metabolic Disorders, Nencki Institute of Experimental Biology, Polish Academy of Sciences, 02-093 Warsaw, Poland; 4Association NBIA Polska, 00-453 Warsaw, Poland

**Keywords:** neurodegeneration with brain iron accumulation, PKAN, MPAN, biomarkers

## Abstract

**Highlights:**

**What are the main findings?**
A correlation between disease severity and serum biomarkers was observed. In MPAN patients, NfL, Tau, and UCH-L1 levels were significantly associated with disease severity, whereas in PKAN patients, Tau, GFAP, and UCH-L1 were the key correlating markers.Despite clinical and pathological similarities between MPAN and PKAN, only MPAN patients showed elevated biomarkers indicating inflammation and blood–brain barrier disruption.

**What is the implication of the main finding?**
Serum biomarkers may serve as useful indicators for monitoring disease progression.The biomarker profile observed in MPAN suggests active neuroinflammation, supporting consideration of anti-inflammatory therapeutic strategies in this subgroup.

**Abstract:**

Background: Biomarker analysis in neurodegeneration with brain iron accumulation (NBIA) can offer valuable insights into the disease’s pathology and natural history. Methods: Twenty-five patients with *C19orf12* mutations causing mitochondrial membrane protein-associated neurodegeneration (MPAN), 12 patients with *PANK2* mutations causing pantothenate kinase-associated neurodegeneration (PKAN), and 30 age- and gender-matched controls were studied. Serum levels of MMP-9, S100B, ICAM-1, E- and P-selectins, total α-synuclein, neurofilament light chain (NfL), glial fibrillary acidic protein (GFAP), Tau, ubiquitin-C-terminal hydrolase-L1 (UCH-L1), and brain-derived neurotrophic factor (BDNF) were measured. Clinical status was evaluated with dedicated rating scales. Results: Compared to the control group, MPAN patients had significantly higher serum levels of nearly all biomarkers, except BDNF. NfL, GFAP, and UCH-L1, were elevated by 5, 2, and 3.5 times, respectively. PKAN patients showed no significant differences in GFAP, UCH-L1, and S100B levels compared to controls. However, NfL and Tau levels were increased by 3 and 1.8 times, respectively. A correlation was observed between disease severity and levels of NfL, Tau, and UCH-L1 in MPAN, and GFAP, Tau, and UCH-L1 in PKAN. Conclusions: Patients with MPAN and PKAN showed increased levels of neurodegeneration biomarkers. Elevated inflammation and blood–brain barrier dysfunction biomarkers were specific to MPAN patients.

## 1. Introduction

In vivo studies of the underlying pathological processes, disease diagnosis, the natural progression of conditions, and potential treatment responses are crucial in clinical practice, although they can be challenging. Recent advances in detection techniques have enabled the reliable measurement of specific brain-derived markers released into the serum in proportion to central nervous system (CNS) damage. These biomarkers have been extensively studied in neurodegenerative and inflammatory diseases, providing valuable insights into the pathological processes and becoming essential for both diagnostic and prognostic purposes. The most extensively studied are neurofilament proteins (Nfs), neuron-specific components responsible for maintaining cell structural integrity. Axonal damage triggers Nfs degeneration, leading to the accumulation of neurofilament light chain protein (NfL). NfL levels are established biomarkers for multiple sclerosis (MS), facilitating the monitoring of subclinical damage, relapse, and disease-modifying therapies [1,2,3]. They are also increased in neurodegenerative diseases such as Parkinson’s disease (PD), Alzheimer’s disease (AD), amyotrophic lateral sclerosis (ALS), frontotemporal dementia (FTD), and atypical Parkinsonism [4,5,6]. Glial fibrillary acidic protein (GFAP) is another important brain protein biomarker that supports the intermediate filament network of astrocytes. GFAP levels are usually elevated in neurodegeneration, especially in conditions associated with significant astrogliosis, such as FTD [7]. Tau, a key member of the microtubule-associated protein family (MAPT), plays a crucial role in maintaining cell structural integrity. Total Tau is a biomarker of neuronal damage. It plays a decisive role in differential diagnoses, especially in AD [8], but also in dementia with Lewy bodies (DLB) or Creutzfeldt–Jakob disease (CJD) [9]. Ubiquitin carboxyl-terminal hydrolase L1 (UCH-L1) is a brain protein that hydrolyzes small C-terminal ubiquitin adducts to produce ubiquitin monomers, serving as a promising new biomarker in neurodegeneration, such as PD or Wilson disease (WD) [10,11].

Neurodegeneration biomarkers are especially interesting in diseases where the underlying pathology is not fully understood, such as neurodegeneration with brain iron accumulation (NBIA). NBIA comprises a heterogeneous group of neurodegenerative disorders characterized by a progressive extrapyramidal syndrome and excessive iron deposition in the basal ganglia, particularly the globus pallidus (GP) and substantia nigra (SN) [12]. The most common NBIA syndrome is pantothenate kinase-associated neurodegeneration (PKAN), which accounts for approximately half of the NBIA cases [13]. It is an autosomal, recessive disorder caused by a mutation in the *PANK2* gene on chromosome 20. *PANK2* encodes the pantothenate kinase enzyme, a key regulatory enzyme in coenzyme A (CoA) biosynthesis [14]. In classical PKAN, the age of onset is typically early childhood, and symptoms are mainly extrapyramidal, with prominent dystonia and pyramidal signs (spasticity, hyperreflexia, Babinski sign) [14]. However, atypical cases with late onset have also been reported. Mitochondrial membrane protein-associated neurodegeneration (MPAN) is the second or third most common NBIA subtype worldwide, caused by autosomal recessive or dominant mutations in the *C19orf12* gene [15,16]. The role of the *C19orf12* protein has not been fully understood, although it plays a role in mitochondrial and lipid metabolism [17]. The average age of MPAN onset is within the first decade of life, although late-onset cases have also been documented. Spasticity and brisk deep tendon reflexes tend to affect the lower limbs earlier and more severely than the upper limbs. Other symptoms include parkinsonism and dystonia, dysarthria, visual impairment with optic nerve atrophy, cognitive decline, or dementia [18,19,20].

In PKAN, the main pathological feature is iron deposition in the globus pallidus, accompanied by the loss of viable neurons, especially in the globus pallidus interna. Axonal spheroids (swollen axons) stain for ubiquitin (larger spheroids), while smaller axonal spheroids test positive for amyloid precursor protein with less ubiquitin staining [21]. Tau pathology is present with neurofibrillary tangles and neuropil threads in various brain regions [22]. Pathological features of MPAN include widely distributed Lewy bodies and Lewy neurites (found in the globus pallidus, striatum, substantia nigra, pons, as well as in the hippocampus and neocortex), iron deposits in the globus pallidus and substantia nigra, axonal impairment (characterized by axonal spheroids with strong ubiquitin immunoreactivity), demyelination, and neuronal loss. To a lesser degree, MPAN also features hyperphosphorylated tau-containing neuronal inclusions, mainly in the hippocampus [15,20].

Considering the neuropathology of NBIA, the aforementioned biomarkers were chosen. Others, less specific but considered interesting, were also picked. Alpha-synuclein is a natural choice for MPAN, as it is a synucleinopathy. Brain-derived neurotrophic factor (BDNF) is low in most neurovegetative diseases, so it was also chosen. S100B protein, a protein associated with neurodegeneration, contributes to increased oxidative stress, which is described in MPAN and PKAN. S100B is also recognized as a biomarker of BBB integrity, so other markers of inflammation and blood–brain barrier integrity, including metalloproteinase 9 (MMP-9), E- and P-selectins, intercellular adhesion molecule (ICAM-1), were also chosen. The main goal of the study was to explore serum biomarkers of neuronal degeneration, the potential role of the blood–brain barrier (BBB) and inflammation in patients with PKAN and MPAN and to identify the most promising candidate for monitoring the disease course and the effectiveness of therapy.

## 2. Materials and Methods

### 2.1. Participants

The study was conducted in the 2nd Department of Neurology at the Institute of Psychiatry and Neurology in Warsaw. It received approval from the local Ethics Committee for Human Research (Nos. 2/2022 and 3/2022). Samples were collected in 2022 from patients and the control group.

Participants had to be at least 12 years old and provide signed informed consent For individuals aged 12–18, permission was granted by both the participant and their legal guardian. Exclusion criteria included lack of genetic confirmation of MPAN/PKAN for the patient group; recent immunosuppressive or immunomodulatory treatment within the past six months; surgery or major trauma within the previous two months; hepatic or renal insufficiency; pregnancy; signs of infection on clinical or laboratory examination; bone fractures within the past six months; severe cardiopulmonary disease; and participation in intense strength training or high-impact workouts within the last week. Each participant was instructed to avoid strenuous exercise for 24 h before testing, as exercise can influence serum biomarker levels.

A neurological exam was conducted on the same day as serum collection, using PKAN-DRS [23] for PKAN patients. For MPAN patients, we employed a scale developed specifically for this disease at our center. The MPAN scale, similar to the PKAN-DRS, includes six subscales that evaluate disease features such as dementia, optic nerve atrophy, or severe spasticity. The detailed subscales are: (I)—activities of daily living, such as feeding, dressing, mobility, self-assessed cognition, and psychiatric symptoms (14 questions); (II)—basic cognitive assessment (including memory and language skills, 6 questions); (III)—dystonia, including oromandibular dystonia and dystonia of the arms and legs (6 questions); (IV)—parkinsonism (6 questions); (V)—spasticity, muscle strength, and tendon reflexes (2 questions); (VI)—other features such as speech, visual acuity, oculomotor function, muscle tone, or gait (7 questions). Each question is scored from 0 (no symptoms) to 4 (severe symptoms).

Blood samples (10 mL) were drawn from fasting participants after a night’s rest and immediately centrifuged. The serum was stored at −80 °C until analysis.

### 2.2. Biomarkers Evaluation

The serum levels of MMP-9, E- and P-selectins, ICAM-1, α-synuclein, BDNF, and S100B were measured using sandwich ELISA according to the manufacturer’s instructions. Kits for ICAM-1, MMP-9, E-selectin, P-selectin, and BDNF from R&D Systems (Minneapolis, MN, USA) were utilized. For S100B and total α-synuclein, a kit from ELK Biotechnology (Wuhan, China) was used. Absorbance at 450 nm was read using a Multiscan Go spectrophotometer (Thermo Scientific, Waltham, MA, USA). Protein concentrations were calculated following the manufacturer’s guidelines. Serum levels of NfL, Tau, GFAP, and UCH-L1 were measured using the single molecule array SR-X analyzer (Quanterix, Boston, MA, USA) with the Neurology 4-Plex B Advantage Kit (Quanterix, Boston, MA, USA). The assay followed the manufacturer’s two-step protocol. Calibrator points were tested in triplicate, and samples along with internal controls were analyzed in duplicates at 1:4 dilution.

### 2.3. Statistical Analysis

Descriptive statistics were used to present baseline characteristics and outcome measurement results, including means, medians, and interquartile ranges (IQRs). Differences in serum biomarkers were assessed using *t*-tests for independent samples, the Mann–Whitney U test for parametric and non-parametric data, and the Pearson χ^2^ test for categorical data. The correlations among biomarker concentrations were evaluated using Spearman’s rank correlation coefficient based on data from all participants. The Statistica 13 software was used.

## 3. Results

### 3.1. Patients

We enrolled 25 patients (9 women and 16 men) with *C19orf12* mutation and 12 (8 women and 4 men) with *PANK2* mutation. Detailed neurological status is shown in Table 1 and Table 2. The MPAN group was younger, with a mean age of 20.8 ± 4.8 years (median: 21 years; range: 14–29 years), compared to the PKAN group, which had a mean age of 25.5 ± 4.9 years (median: 25 years; range: 17–35 years). Since PKAN patients were older than MPAN patients (*p* = 0.0237), we included a larger control group of 30 individuals to match the age distributions of both groups and account for potential age-related differences. Table 3 summarizes the clinical assessment results and all biomarker results.

### 3.2. Biomarkers of Neurodegeneration

To evaluate the differences in levels of neurodegeneration biomarkers, we selected the following: NfL, GFAP, Tau, UCH-L1, alpha-synuclein, and BDNF. As shown in Figure 1 and Table 3, NfL levels were five times higher in the MPAN group and three times higher in the PKAN group compared to the control group. Serum GFAP concentration was twice as high in the MPAN group, while in the PKAN group, it was similar to control levels. One sample from the PKAN group had an extremely high GFAP level and was excluded from the calculation of the mean and standard deviation but included in the statistical analysis. This exclusion did not affect the identification of significant differences. The abnormal sample, tested multiple times, consistently showed a concentration 100 times higher than the group average. It remains unclear whether this was due to a technical issue or if the patient genuinely had an exceptionally high GFAP level. Notably, this patient did not differ significantly from the rest of the PKAN group in other biomarkers. Tau protein levels showed a slight but statistically significant increase in the MPAN group (approximately 10% higher than in controls), and were markedly elevated in the PKAN group (about 80% higher than in controls). UCH-L1 levels were significantly higher only in the MPAN group compared to controls. Similarly, α-synuclein levels were 30 times higher in the MPAN group than in controls. In contrast, the PKAN group showed a modest increase, with levels approximately double those of the control group. BDNF concentration was significantly lower in the PKAN group than in the MPAN group. However, neither of these groups differed significantly from the controls.

### 3.3. Biomarkers of the Blood–Brain Barrier

The potential roles of the blood–brain barrier and inflammation in PKAN and MPAN patients were examined using the following biomarkers: MMP-9, ICAM-1, E- and P-selectin, and S100B. As shown in Figure 2 and Table 3, these biomarkers were significantly elevated in the MPAN group. MMP-9, ICAM-1, E-selectin, P-selectin, and S100 B protein concentrations were 1.7, 1.4, 2, 5.5, and 9 times higher, respectively, compared to the control group. In PKAN patients, both P- and E-selectin levels were markedly elevated, with P-selectin levels 4 times higher than in the control group.

### 3.4. Biomarkers’ Correlations

First, the correlations in the control group were assessed. Levels of E-selectin, BDNF, S100B, NfL, and UCH-L1 increased with age. We observed a positive correlation among inflammatory and BBB biomarkers, including E-selectin, MMP-9, P-selectin, and S100B. Tau concentration showed a moderate correlation with all other degeneration biomarkers, including NfL, GFAP, and UCH-L1. Notably, NfL levels were linked to BDNF and S100B. The UCH-L1 concentration also moderately correlated with E-selectin and S100B (Appendix A). In the MPAN group (Figure 3), a positive correlation was observed between E-selectin and ICAM-1 concentrations, and a moderate correlation between P-selectin and E-selectin. The correlation, along with the significant elevation of these biomarkers, suggests that MPAN patients may experience a chronic CNS inflammatory state. Additionally, these proteins are recognized biomarkers of blood–brain barrier activation and impairment. A strong positive correlation was noted between α-synuclein and S100B concentrations. No correlation was detected between the other neurodegeneration biomarkers—GFAP, NfL, UCH-L1, and Tau—and no significant age-related correlations were observed. The absence of these correlations indicates that high levels of neurodegeneration biomarkers are due to pathological processes rather than inter-dependencies.

In the PKAN group, a strong positive correlation was found between age and MMP9, P-selectin, and BDNF. A strong positive correlation was observed among MMP-9, BDNF, and P-selectin, whereas a strong negative correlation was observed between MMP-9 and S100B. BDNF levels correlated with P-selectin and inversely with S100B, as did NfL. Unlike the MPAN group, GFAP concentration showed a positive correlation with NfL and Tau. Additionally, NfL showed a strong positive correlation with α-synuclein concentration (Figure 3). In contrast to the correlations in the MPAN group, those in the PKAN group were similar to those in the control group.

### 3.5. Correlation of Biomarkers and Clinical Status

In the MPAN group, NfL concentrations showed a strong positive correlation with the first clinical subscale, which assesses daily living, and the fifth subscale, which evaluates pyramidal functions. UCH-L1 correlated with subscale 4, which describes parkinsonism, and with subscale 6, which assesses other functions, including visual acuity, speech, and gait. Tau concentrations were associated with cognition, as indicated by subscale 2 (Figure 3). In the PKAN group, GFAP, Tau, and UCH-L1 demonstrated strong correlations with subscale 5, which evaluates dystonia, and GFAP and Tau also correlated with subscale 6, which includes other neurological signs such as speech, chorea, and spasticity. ICAM-1 concentrations were associated with subscale 1, which measures self-rated cognition in the PKAN group (Figure 3).

## 4. Discussion

Iron deposition, observed in radiological and neuropathological studies of PKAN and MPAN, is unlikely to be the primary cause of the disease [24]. Instead, both conditions involve disruptions in lipid metabolism that may cause dysfunction in iron transport and compartmentalization [25].

We demonstrated that the two main NBIA syndromes, PKAN and MPAN, exhibit distinct serum biomarker profiles compared with each other and with age- and sex-matched healthy controls. Specifically, MPAN patients showed elevated biomarkers of inflammation and BBB dysfunction, whereas no such increases were observed in PKAN patients. MPAN patients also had notably high serum alpha-synuclein levels, consistent with this protein’s accumulation in the brain [15,20]. In contrast, PKAN patients exhibited higher serum Tau protein levels. Both groups had elevated NfL levels, a biomarker of axonal damage.

NfL has become one of the most widely studied biomarkers for inflammatory and neurodegenerative diseases and has become a marker of axonal damage [1,2,3,4,5,6]. In patients with PD, NfL levels are associated with brain atrophy, cognitive decline, or motor impairment and may function as a prognostic indicator for the disease progression [26,27]. Increased NfL concentrations were observed in both MPAN and PKAN, reaching levels similar to those seen in other chronic neurodegenerative diseases [7]. In the MPAN group, NfL levels strongly correlated with clinical evaluations in two assessed areas—pyramidal signs and daily living—clearly indicating the degree of motor function impairment. The higher NfL levels observed in MPAN patients compared to PKAN patients may suggest faster progression of neurodegeneration, more extensive axonal damage at the time of assessment, and more intense demyelination in MPAN than in PKAN. In our study, PKAN patients were older than MPAN patients; therefore, a possible explanation could be that NfL levels decrease with significant neuronal loss in the older group. However, this interpretation remains uncertain and requires further longitudinal studies.

GFAP is a biomarker for both astrogliosis and inflammation. In conditions like FTD, GFAP levels may be higher than those seen in diseases such as AD or PD, and GFAP levels have been shown to correlate with disease severity [7,28]. In the MPAN group, GFAP concentrations were elevated, suggesting more severe astrocyte impairment; however, they did not correlate with clinical disease assessments. Interestingly, although GFAP concentrations were not elevated in the PKAN group, but they did correlate with NfL and Tau levels, as well as evaluations of dystonia and other neurological signs, including speech, spasticity, and chorea, suggesting it could be a biomarker of disease severity.

Microtubule protein Tau correlates with cognitive function and cortical thickness [29]. The serum t-Tau concentration reported in PD [7,8] was similar to that observed in our MPAN and PKAN patients. Additionally, Tau concentration was linked to cognitive impairment in the MPAN group, indicating its possible role as a marker of dementia in this population. In the PKAN group, Tau levels correlated with dystonia and other clinical subscales, including various neurological signs, suggesting its potential as a marker of disease severity. However, it did not correlate with cognitive subscale I of the PKAN-DRS, despite patients reporting cognitive decline.

UCH-L1 is an enzyme that plays a crucial role in cellular metabolism, primarily through its involvement in the ubiquitin–proteasome system, which is responsible for degrading dysfunctional proteins. Additionally, UCH-L1 regulates overall lysosomal activity and cytoskeletal dynamics, acting as a signaling molecule. However, unlike previous biomarkers, it has been less studied and is not linked to a specific neurodegenerative process. In our study, UCH-L1 showed a strong correlation with clinical signs of parkinsonism in MPAN patients and dystonia in PKAN patients, both indicating the severity of disease progression. This biomarker appears essential for long-term monitoring of disease progression.

S100B, alpha-synuclein, and BDNF gave less clear and conclusive results. S100B is expressed in astrocytes, oligodendrocytes, and some neuronal populations, as well as in many cells outside the CNS [30,31]. It is a protein associated with neurodegeneration, contributing to increased oxidative stress, promoting inflammatory cascades, and disrupting Ca^2+^ homeostasis. S100B is also recognized as a biomarker of BBB integrity. Its level depends on: cerebral blood flow, physical activity, and conditions like epileptic seizure [32,33]. BDNF plays a role in mediating synaptic plasticity and neuronal connectivity [34], and its levels are low in neurodegeneration [35] and the aging brain. Our study observed a significant increase in S100B levels in MPAN patients, but not in PKAN patients, and low BDNF levels in both. Since both diseases are neurodegenerative, we suggest that the higher S100B concentration in MPAN patients is more likely related to BBB impairment rather than degeneration.

S100B and BDNF are of interest in MPAN since they are released from various cell types, with adipocytes being the main source [31,36]. Since *C19orf12* is expressed in adipose tissue and plays a crucial role in adipose tissue metabolism [15,37], S100B and BDNF levels in MPAN patients might be associated with impaired lipid metabolism.

ICAM-1, E-selectin, P-selectin, and MMP-9 are soluble proteins secreted by various cell types, especially endothelial cells, platelets, leukocytes, and astrocytes of the blood–brain barrier. These proteins are usually associated with endothelial activation or injury, often as part of processes related to endothelial dysfunction. Although various factors can influence their levels, elevated concentrations are related to inflammatory responses and damage to the blood–brain barrier [32,38,39,40]. Our study excluded conditions known to increase the levels of these proteins. MPAN patients showed significantly higher serum levels of ICAM-1, E-selectin, P-selectin, and MMP-9 compared to healthy controls and PKAN patients. Along with the observed increase in S100B concentrations, these findings suggest that MPAN pathology may induce a chronic inflammatory state and damage the BBB.

To conclude, we emphasize that we have identified reliable serum biomarkers for patients with MPAN and PKAN. NfL is a key biomarker of brain axonal damage and should be measured regularly to track disease progression in MPAN and PKAN. GFAP, UCH-L1, S100B, Tau, and α-synuclein may also be useful for assessing MPAN patients, as they can evaluate disease severity. In PKAN, NfL, UCH-L1, and Tau proteins are the most appropriate biomarkers for monitoring disease progression. Inflammatory and BBB biomarkers are elevated in MPAN patients, unlike in PKAN patients.

A limitation of our study is the small sample size, which could bias the results. Nonetheless, this is a common challenge when researching ultra-rare diseases. Another limitation is the limited confidence in nonspecific biomarkers, such as inflammatory markers, which are affected by many factors. Although our exclusion criteria were designed to minimize confounding variables, eliminating them entirely is impossible. A relatively large spread of individual values is observed; this variability mainly reflects a few outliers within the small study groups. Despite this, the overall data distribution within each group remained approximately normal and consistent, allowing the use of appropriate parametric tests. Therefore, the observed trends—including the direction and relative magnitude of changes in the analyzed molecular markers—remain valid and support the conclusions drawn in the study. Longitudinal studies, preferably involving a larger patient group and more NBIA subtypes, are still necessary to confirm the utility of these biomarkers.

## Figures and Tables

**Figure 1 cells-14-01801-f001:**
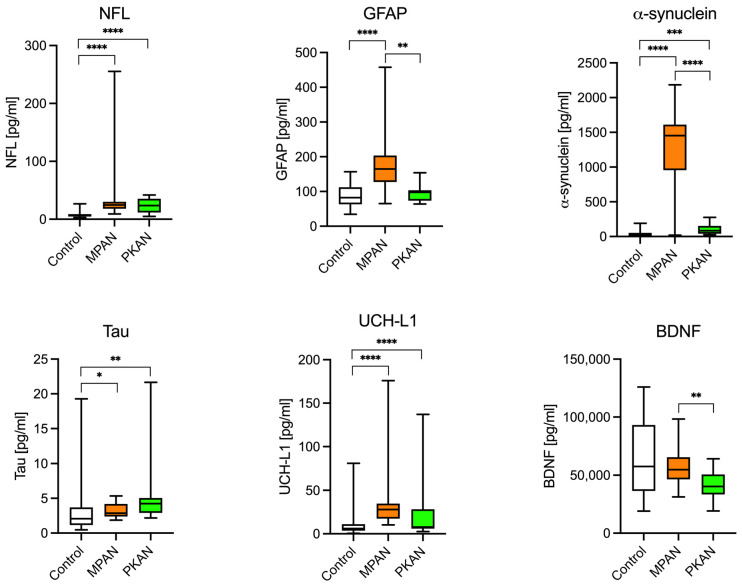
Neurodegeneration biomarkers in the MPAN, PKAN patients, and the control group. NFL (neurofilament light chain protein), α-synuclein, Tau protein, and UCH-L1 (ubiquitin carboxyl-terminal hydrolase L1), well-established markers of neurodegeneration, were significantly elevated in both MPAN and PKAN compared with the control group. Interestingly, GFAP (glial fibrillary acidic protein), a marker of astrogliosis and chronic inflammation, was elevated only in MPAN, not in PKAN. * *p* < 0.05, ** *p* < 0.01, *** *p* < 0.001, **** *p* < 0.0001—*t*-Student test.

**Figure 2 cells-14-01801-f002:**
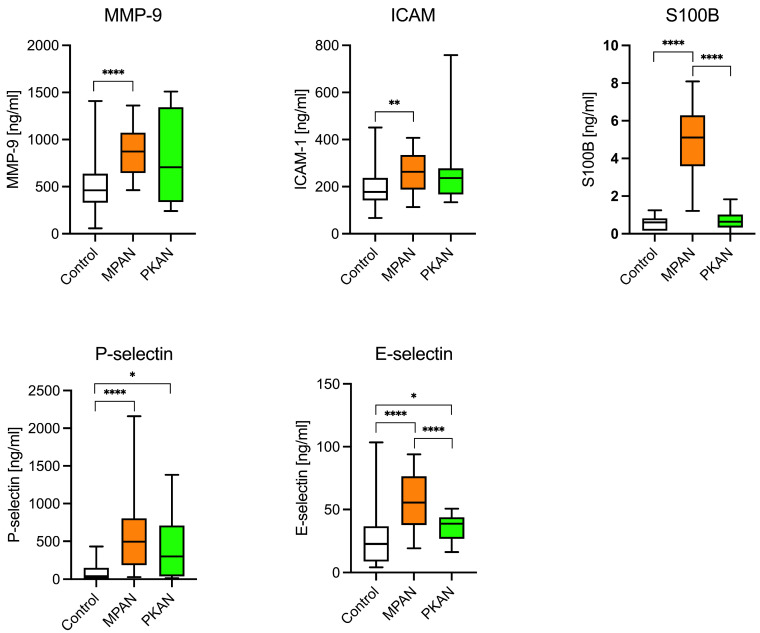
Biomarkers of inflammation and blood-brain-barrier dysfunction in MPAN, PKAN patients, and the control group. The elevation of less-specific biomarkers of neuroinflammation and BBB dysfunction—MMP-9 (metalloproteinase 9), E- and P-selectins, and ICAM-1 (intercellular adhesion molecule)—is typical of MPAN, not of PKAN. * *p* < 0.05, ** *p* <0.01, **** *p* < 0.0001, *t*-Student test.

**Figure 3 cells-14-01801-f003:**
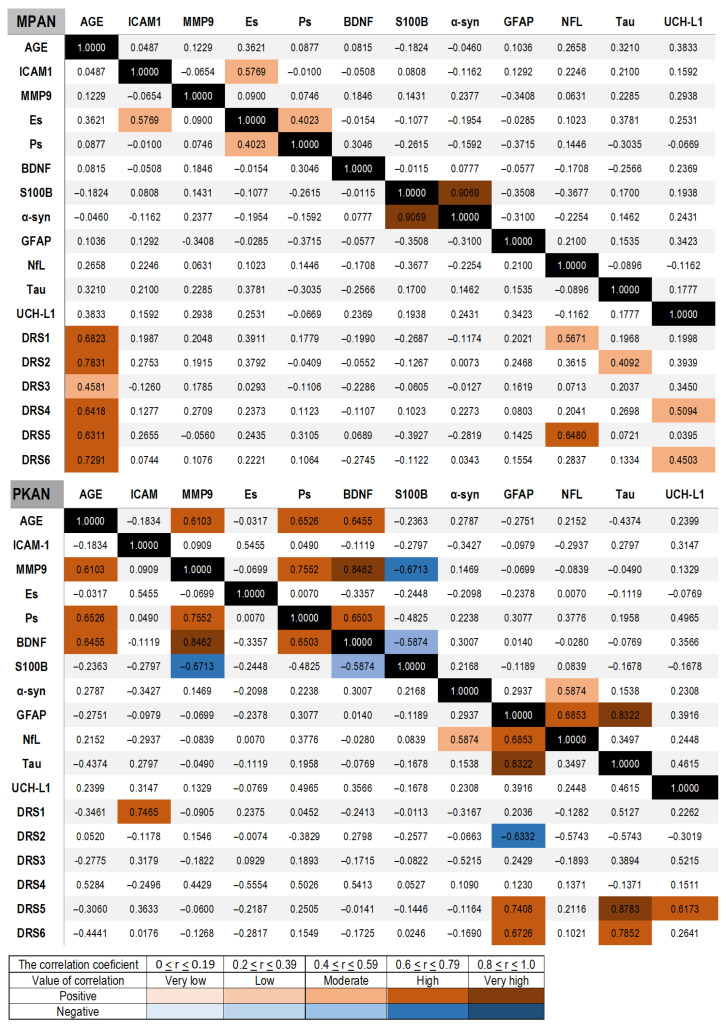
Spearman correlations between serum biomarkers and between clinical evaluations assessed with the MPAN scale and the PKAN-DRS, respectively, in MPAN and PKAN patient groups. The correlation values are shown with color bars, brown for positive and blue for negative. The black is used for the match of the same variable.

**Table 1 cells-14-01801-t001:** Clinical characteristics of PKAN patients.

Patient (Sex)	Mutation	Age of Assessment	Scale I	Scale II	Scale III	Scale IV	Scale V	Scale IV
1. (M)	c. 573delC; 1583C>T; p.(Ser191ArgfsX13); (Thr528Met)	27	2	1	20	10	21	7
2. (M)	c. 573delC; 1274T>C; p.(Ser191ArgfsX13); (Leu425Pro)	28	3	0	27	22	24	12
3. (F)	c. 573delC; 911_913delTCT; p. (Ser191ArgfsX13); (F304del)	28	2	0	27	25	30	11
4. (F)	c. 793G>A; 1203delC; p.(Asp265Asn); (Asp403IlefsX47); c.377G>C; 377G>C; p.(Gly126Ala); (Gly126Ala)-non pat.	29	1	1	27	20	30	13
5. (F)	c. 1561G>A; del exons 3 and 4; p. (Gly521Arg); x	22	2	0	31	16 (+1UR)	28	16
6. (F)	c. 573delC; 863C>G; p.(Ser191ArgfsX13); (Pro288Arg)	28	3	0	24	22	32	13
7. (F)	c. 1583C>T; 1561G>A; p.(Thr528Met); (Gly521Arg)	24	2	2	27	17	27	8
8. (F)	c. 1561G>A; 1561G>A; p.(Gly521Arg); (Gly521Arg)	20	4	1	34	16 (+2UR)	37	16
9. (F)	c.1561G>A; 1561G>A; p.(Gly521Arg); (Gly521Arg)	17	3	0	29	17 (+1UR)	34	14
10. (F)	c. 573delC; 1561G>A; p. (Ser191ArgfsX13); (Gly521Arg)	25	1	3	17	21	19	8
11. (F)	c. 1583C>T; 1561G>A; p. (Thr528Met); (Gly521Arg); c. 377G>C; 377G>C; p.(Gly126Ala); (Gly126Ala)-non pat.	25	2	2	17	14	21	6
12. (F)	c. 573delC; 863C>G; p.(Ser191ArgfsX13); (Pro288Arg)	35	2	2	29	18	21	6

UR—unable to rate. PKAN: pantothenate kinase associated neurodegeneration. Scale I, II, III, IV, V, VI—scales form PKAN-DRS; data presented as total from the scale.

**Table 2 cells-14-01801-t002:** Clinical characteristics of MPAN patients.

Patient (Sex)	Mutation	Age of Assessment	Scale I	Scale II	Scale III	Scale IV	Scale V	Scale VI
1. (F)	H	15	3	4	2	0	6	2
2. (M)	c.32C>T; 204_214del; p.(Thr11Met); (Gly69Argfs*10)	27	27	10	2	16	12	20
3. (M)	H	19	35	11	15	12	8	19
4. (M)	H	24	32	20	16	18	14	45
5. (F)	H	16	16	5	7	14	10	16
6. (F)	c.68T>C; 204_214del11; p.(Leu23Pro); (Gly69Argfs*10)	21	19	14	13	13	15	14
7. (F)	c.68T>C; 204_214del11; p.(Leu23Pro); (Gly69Argfs*10)	21	13	5	10	9	14	12
8. (M)	c.204_214del11; 205G>A; p.(Gly69Argfs*10); (Gly65Arg)	14	21	2	3	2	10	7
9. (F)	H	21	16	6	14	10	10	23
10. (M)	H	19	15	5	17	12	8	19
11. (M)	H	28	24	7	15	14	11	24
12. (M)	H	26	23	9	21	16	12	34
13. (M)	H	24	19	9	13	13	10	29
14. (M)	c.161delG; 204_214del11 p.(Gly54Valfs*19); (Gly69Argfs*10)	15	10	2	6	3	10	14
15. (M)	H	24	35	18	16	16	14	27
16. (M)	H	29	25	15	7	11	14	40
17. (M)	H	19	25	8	19	14	10	29
18. (M)	H	15	9	5	9	15	6	19
19. (F)	c.204_214del11; 424A>G p.(Gly69Argfs*10); (Lys142Glu)	16	8	4	9	9	8	10
20. (F)	H	22	17	12	4	10	8	15
21. (M)	H	26	38	20	25	18	18	43
22. (M)	H	13	14	3	6	9	10	17
23. (F)	c.244_245dup;# p.(Pro83Serfs*7);#	17	22	4	8	7	12	19
24. (F)	c.205G>A; 424A>G; p.(Gly65Glu); (Lys142Glu)	24	36	20	16	20	22	44
25. (M)	H	25	38	15	21	15	14	37

H—homozygotic mutation: c.204_214del; c.204_214del11; p Gly69Argfs*10; p.Gly69Argfs*10. Scale I, II, III, IV, V, VI—scales form the MPAN-scale; data presented as total from the scale. MPAN—mitochondrial membrane protein associated neurodegeneration.

**Table 3 cells-14-01801-t003:** Characteristics of the study group, clinical assessment and biomarker levels in MPAN, PKAN, and control patient groups.

	MPAN	*p* ^1^	PKAN	*p* ^1^	*p* ^2^	Control
Number of patients F/M	25 (9/16)	-	12 (8/4)	-	-	30 (14/16)
Age mean ± SD; years (range)	20.8 ± 4.8 (14–29)	0.16	25.5 ± 4.9 (17–35)	0.7	**0.02**	23.6 ± 4.9 (13–33)
Disease rating scales	MPAN scale			PKAN DRS		
Mean ± SD (range) total	89.1 ± 37.3 (17–162)			82.3 ± 12.8 (61–97)		
Subscale 1	21.6 ± 9.9 (3–38)			2.27 ± 0.9 (1–4)		
Subscale 2	9.3 ± 5.9 (2–20)			1.0 ± 1.1 (0–3)		
Subscale 3	11.7 ± 6.4 (2–25)			26.2 ± 5.2 (17–34)		
Subscale 4	11.4 ± 5.0 (0–20)			19.5 ± 3.4 (14–22)		
Subscale 5	11.4 ± 3.7 (6–22)			27.5 ± 5.8 (19–37)		
Subscale 6	23.1 ± 11.8 (2–44)			11.2 ± 3.7 (6–16)		
Biomarkers	mean ± SD	*p* ^1^	mean ± SD	*p* ^1^	*p* ^2^	mean ± SD
NfL pg/mL	40.11 ± 52.7	**4.3 × 10^−9^**	23.8 ± 12.8	**8.8 × 10^−5^**	7.4 × 10^−1^	7.8 ± 5.1
GFAP pg/mL	175.64 ± 78.6	**5.8 × 10^−8^**	96.7 ± 24.5 *	9.2 × 10^−1^	**8.4 × 10^−3^**	87.6 ± 33.2
Tau pg/mL	3.26 ± 1.14	**2.9 × 10^−2^**	5.4 ± 5.2	**6.6 × 10^−3^**	7.4 × 10^−1^	3.0 ± 3.4
UCH-L1 pg/mL	42.97 ± 57.3	**1.8 × 10^−6^**	25.1 ± 39.8	5.5 × 10^−1^	**5.3 × 10^−5^**	11.9 ± 16.4
MMP-9 ng/mL	874.10 ± 277.7	**1.5 × 10^−5^**	795.5 ± 487.3	1.7 × 10^−1^	0.4	517.5 ± 299.5
ICAM-1 ng/mL	260.20 ± 90.5	**6.3 × 10^−3^**	265.2 ± 164.7	8.7 × 10^−2^	5.8 × 10^−1^	192.8 ± 81.8
E-selectin ng/mL	56.78 ± 22.87	**7.7 × 10^−6^**	36.1 ± 10.9	**2.4 × 10^−2^**	**9.4 × 10^−3^**	28.3 ± 24.9
P-selectin ng/mL	553.70 ± 465.1	**1.8 × 10^−7^**	420.8 ± 454.7	**0.1 × 10^−1^**	2.8 × 10^−1^	100.8 ± 124.7
S100B ng/mL	4.93 ± 1.88	**2.3 × 10^−10^**	0.7 ± 0.5	2.9 × 10^−1^	**4.3 × 10^−9^**	0.5 ± 0.4
BDNF ng/mL	56.53 ± 15.6	0.8	41.99 ± 12.1	7.2 × 10^−2^	**6.9 × 10^−3^**	62.3 ± 31.38
α-synuclein pg/mL	1299.10 ± 518.7	**2.3 × 10^−10^**	104.8 ± 79.2	**5.3 × 10^−4^**	**2.9 × 10^−7^**	40.7 ± 47.5

*p* ^1^ significant difference from the control group; *p* ^2^ significant difference between MPAN and PKAN groups; * one not valid assessment not included. Bold is for significant *p*.

## Data Availability

Raw data were collected at the Second Department of Neurology, Institute of Psychiatry and Neurology, in Warsaw, Poland, and at the Laboratory of Neuroimmunology within the same department. Supporting data for this study’s findings are available from the corresponding author upon request.

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
