# Peer review of "Distinct Neurodegenerative Pathways in Two NBIA Subtypes: Inflammatory Activation in C19orf12 but Not in PANK2 Mutation Carriers"

_cells, 2025, doi:10.3390/cells14221801_

Round 1
Reviewer 1 Report
Comments and Suggestions for Authors
Review comments are attached

Review comments are attached
Author Response
Dear Reviewer,
Thank you very much for all your questions and concerns. We have addressed them in the manuscript, hoping it will improve the paper. Please find the answers below.
Reviewer #1
The manuscript describes in detail a study on biomarker analysis in neurodegeneration, based on brain iron accumulation (NBIA). The specific thesis can offer valuable insight into neurodegenerative pathologies, with the associated and disassociated factors emerging as determining vehicles for monitoring neurodegeneration (severity, progress, etc.), thereby laying the groundwork for potential diagnostic and therapeutic approaches. Mitochondrial membrane protein-associated neurodegeneration (MPAN) and pantothenate kinase-associated neurodegeneration (PKAN) pathologies were selected for the specific investigation, with several molecular markers playing distinctly differentiated roles, finally exemplifying relations relevant to symptomatic evidence in real life neurodegenerative condition patients. The work was conducted on small groups of patients (a clear limitation), chosen under strict criteria to meet the demands of the experimental survey, along with the appropriate control group. The results reflect trends in specific molecular markers (from the list of those chosen and actually used), with useful suggestions for their use in future work with such patients. The specific research deserves attention, yet several points need to be delineated by the authors prior to any consideration. A selection of such points is provided below and remarked upon.
- Comment: In the abstract, the statement “Clical status was evaluated with dedicated rating scales.” needs to be rewritten to read “Clinical status was evaluated with dedicated rating scales.”.
Response: We agree with this comment – spelling mistake. The change was made in the Abstract section
- Comment: In the introduction section, lines 80-83, the statement “NBIA comprises a heterogeneous group of neurodegenerative disorders that present with a progressive extrapyramidal syndrome and excessive iron deposition in the basal ganglia, mainly the globus pallidus (GP) and substantia nigra (SN) [12].” is not easily understood. It appears that it should be rewritten to read “NBIA comprises a heterogeneous group of neurodegenerative disorders linked to a progressive extrapyramidal syndrome and excessive iron deposition in the basal ganglia, mainly the globus pallidus (GP) and substantia nigra (SN) [12].”.
Response: We agree with this comment. The text was rephrased for more clarity in the Introduction section
- Comment: In the Materials and methods section, lines 126-127, the statement “Participants had to be at least 12 years old and provide signed informed consent. For individuals aged 12– 18, both the participant and their legal guardian obtained permission.” does not sound right. Do the authors mean “Participants had to be at least 12 years old and provide signed informed consent. For individuals aged 12–18, permission was granted by both the participant and their legal guardian.”?
Response: We agree with this comment. The text was rephrased for more clarity in the Materials section
- Comment: In Table 3, which presents the characteristics of the study group, clinical assessment and biomarker levels in MPAN, PKAN, and control patient groups, the numerical values and the provided-associated standard deviations (in each case presented) are not compatible with the proper number of significant figures. The authors are advised to look into them and provide the numerical data on MMPAN, PKAN, p1, and control group with the appropriate number of significant figures, where appropriate.
Response: We agree with this comment. Table 3 was changed, as you suggested
- Comment: In figures 1 and 2, where graphs are provided on the variation of the molecular factors investigated in both MPAN, PKAN, and control groups, one cannot help but wonder about the spread of values resulting in the statistically significant or insignificant changes that occurred compared to the chosen controls. A statement should be provided, linking that aspect to the derived conclusions on the trends discussed about the specific molecular markers studied.
Response: Although a relatively large spread of individual values can be observed in Figures 1 and 2, this variability mainly reflects the presence of a few outliers within the relatively small study groups. Despite this, the overall data distribution within each group remained approximately normal and consistent, allowing the use of appropriate parametric tests. Therefore, the observed trends — including the direction and relative magnitude of changes in the analyzed molecular markers — remain valid and support the conclusions drawn in the study. We add this explanation as another limitation of the study.
- Comment: The case of NBIA-associated syndromes MPAN and PKAN are clear. However, in the introduction section, there is no clear distinction between NBIA and non-NBIA potential cases that justified the specific choice made for the study of the specific molecular markers selected.
Response: We agree with this comment. The Introduction part was corrected, and the choice of these markers was explained.
- Comment: The authors correctly point out the fact that the size of the groups studied is limited, thereby presenting possibilities for bias in the study results. To that end, there are two points that should be stressed concurrently and inter-dependently. These include both size and longitudinal studies into the factors addressed in the specific study. In that respect, a statement should be added in the conclusions, emphasizing the interdependence of the actions on those points taken in the future, with the caution to be exercised on the results extending to larger population groups in future studies.
Response: We agree with this comment. The limitations were outlined in the Discussion section
Based on the aforementioned grounds, the manuscript should be revised and submitted for evaluation. 2

Reviewer 2 Report
Comments and Suggestions for Authors
Skowronska et al. report on two cohorts of patients with “neurodegeneration with brain iron accumulation” (NBIA), a family of relatively rare neurological diseases caused by, e.g., mitochondrial damage (MPAN, e.g. by c19orf12) or pantothenate kinase mutations (PKAN). They measured various biomarkers in blood and correlated the results with the neurological status. Although the study was relatively small (Tab. 1 and 2), they found several useful associations (Table 3 and Fig. 3). The group has previously published good work on neurodegenerative disease. The present manuscript is well written and offers new insight into a family of rare neurological diseases.
Comments:
- Tables should be understandable by itself without consulting the text. Table legends should be more precise, e.g. with respect to statistics in Table 3. In Table 3, please avoid mixed upper cases and simplify numbers, e.g. 0.0043 x 10-6 equals 4.3 x 10-9. Actually, very low values should be listed as p<0.001. p values below 10-6 lack significance and, given the large SD values, raise doubts.
- In the legend of the Figures, please add explanations. For instance, it is unclear what is shown in Figs. 1 and 2 – quartiles ? What does *** or **** signify? Maybe some y-axes should be logarithmic to accentuate differences.
- The discussion is well written but a rearrangement would benefit the reader. In the present version, one biomarker after another is discussed which makes boring reading. It would be easier for the reader if the biomarkers were discussed together when they have similar significance, e.g. biomarkers that concern neurons (axons, mitochondria) vs. those that reflect glial activation vs. those that bear on blood-brain barrier dysfunction. Also, those biomarkers that are changed significantly deserve more extensive discussion than those that show no differences.
Author Response
Dear Reviewer,
Thank you very much for all your questions and concerns. We have addressed them in the manuscript, hoping it will improve the paper. Please find the answers below.
Reviewer #2
Comments and Suggestions for Authors
Skowronska et al. report on two cohorts of patients with “neurodegeneration with brain iron accumulation” (NBIA), a family of relatively rare neurological diseases caused by, e.g., mitochondrial damage (MPAN, e.g. by c19orf12) or pantothenate kinase mutations (PKAN). They measured various biomarkers in blood and correlated the results with the neurological status. Although the study was relatively small (Tab. 1 and 2), they found several useful associations (Table 3 and Fig. 3). The group has previously published good work on neurodegenerative disease. The present manuscript is well written and offers new insight into a family of rare neurological diseases.
Comments:
- Comment: Tables should be understandable by itself without consulting the text. Table legends should be more precise, e.g. with respect to statistics in Table 3. In Table 3, please avoid mixed upper cases and simplify numbers, e.g. 0.0043 x 10-6 equals 4.3 x 10-9. Actually, very low values should be listed as p<0.001. p values below 10-6 lack significance and, given the large SD values, raise doubts.
Response: We agree with this comment. Table 3 was changed, as you suggested
- Comment: In the legend of the Figures, please add explanations. For instance, it is unclear what is shown in Figs. 1 and 2 – quartiles ? What does *** or **** signify? Maybe some y-axes should be logarithmic to accentuate differences.
Response: We agree with this comment. The Figure legends for Figures 1 and 2 were clarified.
- Comment: The discussion is well written but a rearrangement would benefit the reader. In the present version, one biomarker after another is discussed which makes boring reading. It would be easier for the reader if the biomarkers were discussed together when they have similar significance, e.g. biomarkers that concern neurons (axons, mitochondria) vs. those that reflect glial activation vs. those that bear on blood-brain barrier dysfunction. Also, those biomarkers that are changed significantly deserve more extensive discussion than those that show no differences.
Response: We agree with this comment. We did our best to make the discussion more “fluent” and incorporated the changes you suggested. Since each biomarker is associated with different neuropathological processes, it is impossible to group them together.

Round 2
Reviewer 1 Report
Comments and Suggestions for Authors
The revised version of the manuscript is now acceptable.
Comments on the Quality of English LanguageThe revised version of the manuscript is now acceptable.